# Novel Copper(II) Complexes with Dipinodiazafluorene Ligands: Synthesis, Structure, Magnetic and Catalytic Properties

**DOI:** 10.3390/molecules27134072

**Published:** 2022-06-24

**Authors:** Iakov S. Fomenko, Medhanie Afewerki, Marko I. Gongola, Eugene S. Vasilyev, Lidia S. Shul’pina, Nikolay S. Ikonnikov, Georgiy B. Shul’pin, Denis G. Samsonenko, Vadim V. Yanshole, Vladimir A. Nadolinny, Alexander N. Lavrov, Alexey V. Tkachev, Artem L. Gushchin

**Affiliations:** 1Nikolaev Institute of Inorganic Chemistry, Siberian Branch of Russian Academy of Sciences, 3 Acad. Lavrentiev Ave., 630090 Novosibirsk, Russia; fom1-93@mail.ru (I.S.F.); denis@niic.nsc.ru (D.G.S.); spectr@niic.nsc.ru (V.A.N.); lavrov@niic.nsc.ru (A.N.L.); 2Department of Natural Sciences, Novosibirsk State University, 1 Pirogova Str., 630090 Novosibirsk, Russia; medahaben@gmail.com (M.A.); m.gongola@g.nsu.ru (M.I.G.); 3Vorozhtsov Novosibirsk Institute of Organic Chemistry, Siberian Branch of Russian Academy of Sciences, 9 Acad. Lavrentiev Ave., 630090 Novosibirsk, Russia; vasilyev@nioch.nsc.ru (E.S.V.); atkachev@nioch.nsc.ru (A.V.T.); 4A.N. Nesmeyanov Institute of Organoelement Compounds, Russian Academy of Sciences, ul. Vavilova. 28, 119991 Moscow, Russia; shulpina@ineos.ac.ru (L.S.S.); ikonns@ineos.ac.ru (N.S.I.); 5N.N. Semenov Federal Research Center for Chemical Physics, Russian Academy of Sciences, ul. Kosygina, 4, 119991 Moscow, Russia; gbsh@mail.ru; 6Chemistry and Physics, Plekhanov Russian University of Economics, Stremyannyi Pereulok, 36, 117997 Moscow, Russia; 7International Tomography Center, Siberian Branch of Russian Academy of Sciences, 3a Institutskaya Str., 630090 Novosibirsk, Russia; vadim.yanshole@tomo.nsc.ru

**Keywords:** copper(II), complexes, dipinodiazafluorenes, synthesis, crystal structure, magnetic properties, EPR, catalysis, oxidation, alkanes, alcohols, hydrogen peroxide, tert-butyl hydroperoxide

## Abstract

The reactions of CuX_2_ (X = Cl, Br) with dipinodiazafluorenes yielded four new complexes [CuX_2_L_1_]_2_ (X = Cl (**1**), Br (**2**), L_1_ = (1R,3R,8R,10R)-2,2,9,9-Tetramethyl-3,4,7,8,9,10-hexahydro-1H-1,3:8,10-dimethanocyclopenta [1,2-b:5,4-b’]diquinolin-12(2H)-one) and [(CuX_2_)_2_L_2_]n (X = Cl (**3**), Br (**4**), L_2_ = (1R,3R,8R,10R,1’R,3’R,8’R,10’R)-2,2,2’,2’,9,9,9’,9’-Octamethyl-1,1’,2,2’,3,3’,4,4’,7,7’,8,8’,9,9’,10,10’-hexadecahydro-1,3:1’,3’:8,10:8’,10’-tetramethano-12,12’-bi(cyclopenta [1,2-b:5,4-b’]diquinolinylidene). The complexes were characterized by IR and EPR spectroscopy, HR-ESI-MS and elemental analysis. The crystal structures of compounds **1**, **2** and **4** were determined by X-ray diffraction (XRD) analysis. Complexes **1**–**2** have a monomeric structure, while complex **4** has a polymeric structure due to additional coordinating N,N sites in L_2_. All complexes contain a binuclear fragment {Cu_2_(μ-X)_2×2_} (X = Cl, Br) in their structures. Each copper atom has a distorted square-pyramidal coordination environment formed by two nitrogen atoms and three halogen atoms. The Cu-N_ax_ distance is elongated compared to Cu-N_eq_. The EPR spectra of compounds **1**–**4** in CH_3_CN confirm their paramagnetic nature due to the d^9^ electronic configuration of the copper(II) ion. The magnetic properties of all compounds were studied by the method of static magnetic susceptibility. For complexes **1** and **2**, the effective magnetic moments are *µ*_eff_ ≈ 1.87 and 1.83 *µ*_B_ (per each Cu^2+^ ion), respectively, in the temperature range 50–300 K, which are close to the theoretical spin value (1.73 *µ*_B_). Ferromagnetic exchange interactions between Cu(II) ions inside {Cu_2_(μ-X)_2_X_2_} (X = Cl, Br) dimers (*J*/*k*_B_ ≈ 25 and 31 K for **1** and **2**, respectively) or between dimers (*θ*′ ≈ 0.30 and 0.47 K for **1** and **2**, respectively) were found at low temperatures. For compounds **3** and **4**, the magnetic susceptibility is well described by the Curie–Weiss law in the temperature range 1.77–300 K with *µ*_eff_ ≈ 1.72 and 1.70 *µ*_B_ for **3** and **4**, respectively, and weak antiferromagnetic interactions *(θ* ≈ −0.4 K for **3** and −0.65 K for **4**). Complexes **1**–**4** exhibit high catalytic activity in the oxidation of alkanes and alcohols with peroxides. The maximum yield of cyclohexane oxidation products reached 50% (complex **3**). Based on the data on the study of regio- and bond-selectivity, it was concluded that hydroxyl radicals play a decisive role in the oxidation reaction. The initial products in reactions with alkanes are alkyl hydroperoxides.

## 1. Introduction

Chiral pyridines are of great interest from many points of view [1] and are being actively investigated as ligands in coordination chemistry [2,3,4,5], as chiral auxiliary [6,7,8,9,10,11,12], as biologically active compounds [13,14], including anticancer agents [15], as building blocks in the design of receptors for enantioselective recognition [16], chiral coordination frameworks [17], luminescent materials [18,19], and calamitic liquid crystals [20]. The coupling of a pinane carbon frame and pyridine ring system in one molecule generates an extensive group of pinane–pyridine hybrids (pinopyridines) [21,22,23,24,25,26,27,28,29,30,31,32,33]. Primary functionalization of natural monoterpene hydrocarbon α-pinene by nitrosochlorination-dehydrochlorination leads to pinocravone oxime, which is a convenient starting compound in the synthesis of various pinopyridines, including *C*_2_-symmetric dipinopyridines [34,35,36], which can be considered as derivatives of 2,2′-bipyridine. Molecules of this type are of special interest because they provide coordination to various transition metals [37,38]. Two pinopyridine molecules can be joined through pyridine fragments to form derivatives containing a 4,5-diazafluorene core. 4,5-Diazafluorene is known to be a prospective bidentate ligand providing the formation of transition metal complexes demonstrating a variety of useful properties [19,39,40,41,42]. Certain derivatives of 4,5-diazafluorene belong to the group of the so-called bistricyclic aromatic enes (BAEs). BAEs and related polycyclic systems are a class of molecular materials that display a rich variety of conformations, dynamic stereochemistry and switchable chirality, color, and spectroscopic properties [43,44,45]. The chelating fragment of 4,5-diazafluorene incorporated into an overcrowded alkene molecule leads to new, unusual properties of the transition metal complexes [42,46].

The combination of two pinopyridine fragments forming a diazafluorene core leads to the formation of a chiral dipinodiazafluorene structure, which appears to be a promising chelating ligand for coordination chemistry. On the other hand, it is well known that many complexes of transition [47,48,49,50,51,52,53,54,55,56,57,58,59,60,61,62,63,64,65,66,67,68,69] and non-transition metals [70,71,72,73] catalyze oxidation reactions with peroxides (H_2_O_2_; tert-butyl hydroperoxide), and the use of N-donor chelating ligands in many cases leads to an increase in catalytic activity [74]. In particular, copper complexes with N-donor ligands belong to the effective catalysts for the oxidation of organic compounds [75,76,77,78,79]. In this work, we report on the complexing properties of chiral derivatives of dipinodiazafluorene with respect to copper(II) halides, the preparation of four new copper(II) complexes, and their catalytic activity in the oxidation of alkanes and alcohols with hydrogen peroxide and *tert*-butyl hydroperoxide.

## 2. Results and Discussion

### 2.1. Synthesis

Complexes **1**–**4** were obtained by similar synthetic procedures (Figure 1), by mixing solutions of ligands L_1_ ((1R,3R,8R,10R)-2,2,9,9-Tetramethyl-3,4,7,8,9,10-hexahydro-1H-1,3:8,10-dimethanocyclopenta [1,2-b:5,4-b’]diquinolin-12(2H)-one) or L_2_ ((1R,3R,8R,10R,1’R,3’R,8’R,10’R)-2,2,2’,2’,9,9,9’,9’-Octamethyl-1,1’,2,2’,3,3’,4,4’,7,7’,8,8’,9,9’,10,10’-hexadecahydro-1,3:1’,3’:8,10:8’,10’-tetramethano-12,12’-bi(cyclopenta [1,2-b:5,4-b’]diquinolinylidene)) in chloroform with solutions of CuX_2_ in methanol (CuCl_2_) or acetonitrile (CuBr_2_) and further stirring the mixture for 1 h with slight heating (**1** and **2**) or at room temperature (**3** and **4**). Fine-crystalline powders of complexes **1**–**4** were obtained by evaporation of the resulting solutions in yields from 55 to 74%. Complexes **1** and **2** were highly soluble in standard organic solvents such as dichloromethane and acetonitrile, while complexes **3** and **4** had lower solubility due to the polymer structure (see below).

### 2.2. Crystal Structures

Single crystals of complexes **1**–**2** suitable for X-ray diffraction analysis were obtained by layering diethyl ether on solutions of complexes in dichloromethane. Single crystals of complex **4** were obtained by slow evaporation of the reaction solution. We failed to obtain single crystals of complex **3** suitable for X-ray diffraction analysis. Presumably, complexes **3** and **4** have a similar crystal structure due to the similarity of their composition and properties. The structures of **1**, **2** and **4** are shown in Figure 1. Complexes **1** and **2** have an island structure built by binuclear fragments {Cu_2_(μ-X)_2_X_2_}, in which copper atoms are additionally bonded to L_1_ chelating ligands. The binuclear unit {Cu_2_(μ-X)_2_X_2_} can also be distinguished in structure **4**, but the presence of additional coordination centers in L_2_ leads to a polymeric structure. The CuX_3_N_2_ coordination site in **1**, **2**, and **4** has a distorted square-pyramidal structure. The equatorial plane consists of two bridged μ-X atoms, a terminal X atom, and one N atom of the diimine ligand. The axial position is occupied by the second N atom of the diimine ligand.

Selected bond lengths and angles for **1**, **2** and **4** are given in Table 1. The average Cu-X(terminal) bond lengths are 2.2124(8)–2.2262(8) Å for **1** and 2.3543(8)–2.3735(8) Å for **2**, while the Cu-X(bridge) bond lengths are 2.3069 Å for **1** and 2.4407 Å for **2**. For complex **4**, the average Cu-Br(terminal) and Cu-Br(bridging) distances are 2.3914(10) and 2.3847(8)–2.4988(9) Å, respectively. In all complexes, there is a very strong asymmetry in the coordination of the diimine ligand. The Cu-N(equatorial) bond lengths are 2.013(3)–2.040(3) Å for **1**, 2.003(4)–2.033(4) Å for **2**, and 1.989(5) Å for **4**, while the Cu-N(axial) distances are 2.503(2)–2.595(2) Å for **1**, 2.459(3)–2.543(4) Å for **2**, and 2.432(5) Å for **4**.

The compounds **1** and **2** are isostructural. The binuclear complexes are situated in general positions. The complexes are stacked along the *b*-axis to form columns. The columns alternate along the *a*-axis to form layers parallel to the *ab* plane. The layers are stacked along the *c*-axis to form a mono-layered crystal packing. There are no specific interactions between the binuclear complexes besides van der Waals ones. In the **1** structure, the shortest intermolecular contacts are CH…HC 2.16 Å, CH…C 2.86 Å, CH…Cl 2.85 Å, and CH…OC 2.63 Å. In the **2** structure, the shortest intermolecular contacts are CH…HC 2.20 Å, CH…C 2.90 Å, CH…Br 2.97 Å, and CH…OC 2.52 Å.

The binuclear {Cu_2_(μ-Br)_2_Br_2_} units in **4** are interconnected via bridging organic linkers to form Zigzag chains parallel to the *c*-axis. The chains alternate along the [110] direction to form a layer parallel to the (110) plane. The layers are stacked along the [110] direction to form a mono-layered crystal packing. There are no specific interactions between the binuclear complexes besides van der Waals ones. The shortest intermolecular contacts are CH…HC 2.19 Å, CH…C 2.82 Å, CH…Br 2.97 Å. The structure has isolated voids (26%), which are filled by guest CHCl_3_ molecules.

### 2.3. HR-ESI-MS Studies

High-resolution electrospray mass spectra (HR-ESI-MS) were recorded for complexes **1**–**4** in acetonitrile solution. The electrospray ionization of complexes led to the formation of a number of ion species, formed by fragmentation, formation of adducts with solvent molecules and associates with cations, and the combination of thereof. The mass spectra of complexes **1** and **2** (Appendix A, Appendix A) show a similar pattern for [Cu_2_(μ-X)_2_X_2_}(L_1_)_2_] (X = Cl(**1**), Br(**2**)). In both cases, signals from associates {L_1_+Cat}^+^ and {(L_1_)_2_+Cat}^+^ of the free ligand with H^+^, Na^+^, K^+^, and Cu^+^ cations, as well as signals from mononuclear fragments {Cu(L_1_)(H_2_O)}^+^ (m/z = 451.144) and {Cu(L_1_)(CH_3_CN)}^+^ (m/z = 474.161) were found. In the case of complex **2**, bromide containing form {Cu(L_1_)Br}^+^ (m/z = 512.052) was also detected. In addition, peaks from {Cu(L_1_)_2_}^+^ (m/z = 803.340) for **1** and {Cu(L_1_)_2_Br}^+^ (m/z = 882.256) for **2** were registered. In addition to signals from mononuclear fragments, peaks from binuclear species {Cu_2_(L_1_)_2_Cl}^+^ (m/z = 901.236), {Cu_2_(L_1_)_2_Cl_2_}^+^ (m/z =936.205) for **1** and {Cu_2_(L_1_)_2_Br}^+^ (m/z = 945.186), {Cu_2_(L_1_)_2_Br_2_}^+^ (m/z = 1024.104) for **2** were found. A peak at m/z = 999.131 for **1** was assigned to the trinuclear unit {Cu_3_(L_1_)_2_Cl_2_}^+^.

Fragmentation of complexes **3** and **4** in acetonitrile is slightly different (Appendix A, Appendix A). For both complexes **3** and **4**, peaks from {Cu_2_(L_2_)(CH_3_CN)_2_}^2+^ (m/z = 458.166) and {Cu(L_2_)_2_}^+^ (m/z = 1479.767) were found. Signals from {Cu(L_2_)_2_+Na}^+^ (m/z = 751.389), {Cu(L_2_)(CH_3_CN)}^+^ (m/z = 812.375) were also detected for **3**. The signals from {Cu_2_(L_2_)_2_(CH_3_CN)}^2+^ (m/z = 791.861) and {Cu(L_2_)_3_}^+^ (m/z = 2188.193) were found for **4**. Peaks from halogen containing forms {Cu_2_(L_2_)_2_Br}^+^ (m/z = 1621.616) and {Cu_2_(L_2_)_2_Br_2_}^+^ (m/z = 1700.534) were detected for **4**.

Thus, the mass spectral data indicate complex fragmentation and aggregation in solutions of complexes **1**–**4** (under the conditions of spraying in an electric field), which is consistent with the high kinetic lability of Cu(II) compounds. Molecular complexes **1**–**2** are not stable in acetonitrile under these conditions, and the binuclear{Cu_2_X_4_} (X = Cl, Br) unit undergoes partial destruction with the formation of mononuclear species and associates. The polymeric structure of complexes **3** and **4** is destroyed due to the coordination of acetonitrile molecules to produce mono- and binuclear species.

### 2.4. EPR Spectroscopy Studies

The paramagnetic nature of the Cu(II) complexes was confirmed by EPR spectroscopy data. The EPR spectra of frozen solutions of monomeric complexes **1** and **2** in acetonitrile are shown in Figure 2. The EPR spectrum of complex **1** is a superposition of a wide unresolved line with g = 2.123 and a weakly resolved spectrum with the spin Hamiltonian parameters: g_zz_ = 2.27, g_xx_ = g_yy_ = 2.05 and A_zz_ = 15.8 mT, A_xx_ = A_yy_ = 2.0 mT (Figure 2a). The presence of two spectra with similar average g-factors may be explained by the fact that, due to the extremely low solubility of the complex in acetonitrile at 77 K, part of the complex separates from the solution as a crystalline phase. On the contrary, the EPR spectrum of complex **2** is described by the spin Hamiltonian with the following parameters: g_xx_ = 2.110, g_yy_ = 2.005, g_zz_ = 2.250, A_xx_ = 7.8 mT, A_yy_ = 3.0 mT, A_zz_ = 15.0 mT and a hyperfine structure from the bromine atom along the z direction: A_zz_(Br) = 3.4 mT (Figure 2b).

Due to the low solubility of polymeric complexes **3** and **4** in common solvents, their EPR spectra were not recorded. Instead, the EPR spectra of solutions of reaction mixtures prior to isolation of the crystalline products **3** or **4** were recorded. These solutions were assumed to contain monomeric species similar to **1** and **2**, which subsequently polymerized into structures **3** and **4**. The EPR spectra of crystalline products **1** or **2** were uninformative due to exchange interactions between copper(II) ions (see below). The EPR spectra of the reaction solutions before isolation of **3** or **4** at 77 are shown in Figure 3.

In the case of complex **3**, two well-resolved EPR spectra of copper complexes with different g-factors and HFS constants were overlapped (Figure 3a). The EPR spectra were modeled by a spin Hamiltonian with parameters g_zz_ = 2.225, g_xx_ = g_yy_ = 2.0898, A_zz_ = 12.5 mT, A_xx_ = A_yy_ ~ 0.5 mT (Figure 3a, blue line) and g_zz_ = 2.4208, g_xx_ = g_yy_ = 2.0835, A_zz_ = 11.3 mT, A_xx_ = A_yy_ ~ 0.5 mT (Figure 3a, black line). Both EPR spectra are characteristic of copper(II) ions in an octahedral environment with tetragonal distortion. The values of g-factors and HFS constants of the spectrum in Figure 3a (black line) are typical for the oxygen environment of the Cu(II) ion [80,81]. This may indicate the coordination of water to the copper ion (copper chloride hydrate may be the source of water) to form [Cu_2_(μ-Cl)_2_Cl_2_(H_2_O)_2_(L_2_)_2_]. A decrease in the g_zz_ factor and an increase in the Azz constant for the spectrum in Figure 3a (blue line) means that the ligands of the nearest environment are coordinated to copper ions by atoms with smaller spin–orbit interaction constants compared to oxygen, for example, by nitrogen atoms. This may correspond to the coordination of acetonitrile in [Cu_2_(μ-Cl)_2_Cl_2_(CH_3_CN)_2_(L_2_)_2_]. Similar effects were detected for niobium(IV) halide complexes with d^1^ configuration [82]. The EPR spectrum of **4** is well described by the spin Hamiltonian Ĥ = gβHŜ + A(Cu)ŜÎ + A(Br)ŜÎ with the parameters given below: A(Cu)_zz_ = 11.0 mT, A_xx_ = 5 mT, A_yy_ = 3 mT; g_zz_ = 2.20, g_xx_ = 2.112, g_yy_ = 2.04, A(Br)_xx_ = 3 mT (Figure 3b). The absence of allowed spectrum components to a greater extent corresponds to partially polymerized fragments of compound **4**.

### 2.5. Magnetic Measurements

For complexes **1** and **2**, the measured magnetic susceptibility demonstrated a paramagnetic behavior without any anomaly or magneto-thermal irreversibility (Figure 4a), pointing to the absence of long-range magnetic ordering down to the lowest accessible temperature of 1.77 K. In the temperature range 50–300 K, the magnetic susceptibility can formally be described by the Curie–Weiss law $\chi_{p}\left(T\right)=N_{A}\mu_{\mathrm{eff}}^{2}/3k_{B}\left(T-\theta \right)$ with µ_eff_ ≈ 1.87 µ_B_ and 1.83 µ_B_, θ ≈ 5.5 K and 6.4 K for complexes **1** and **2**, respectively (Figure 4b). The obtained µ_eff_ values are close to those expected for Cu^2+^ (*S* = 1/2) ions and observed in EPR experiments, while positive θ should indicate the ferromagnetic (FM) type of exchange interactions between Cu^2+^ ions. However, given that both complexes do not order ferromagnetically down to 1.77 K, that is, much lower than θ, the evaluated θ values should be considered as an indication of local FM interactions rather than long-range ones. Indeed, according to the established crystal structures (Figure 1), complexes **1** and **2** contain binuclear fragments {Cu_2_(μ-X)_2_X_2_} (X = Cl, Br) separated from each other. Hence, a simple Curie–Weiss description is not really appropriate for the magnetic system of complexes **1** and **2**, and the one focused on pairs of Cu^2+^ ions interacting through exchange forces [83] should be used instead.

Temperature dependences of the effective moment μ_eff_ per Cu^2+^ ion calculated for the case of noninteracting magnetic moments (θ = 0) illustrate the evolution of the magnetic state of Cu^2+^ dimers (Figure 4b). At room temperature, μ_eff_ is close to that of a single Cu^2+^ ion, implying the paramagnetic state of all ions. Upon cooling below ~100 K, μ_eff_ increases, reflecting a gradual formation of a triplet state in each Cu^2+^ dimer. It is worth noting that the growth of effective moments clearly exceeds the coefficient of ≈1.155 theoretically predicted for isolated FM pairs of S = 1/2 ions. This gives unambiguous evidence for the presence of an additional weak FM exchange interaction between Cu^2+^ dimers besides the FM coupling inside them. A good fit to the data has been obtained by using the Bleaney–Bowers equation [83] modified to account for the inter-pair interaction $\chi_{p}\left(T\right)=[N_{A}g^{2}\mu_{B}^{2}/k_{B}\left(T-\;\theta^{\prime }\right)]\left[1/\left(3+\exp(-J/k_{B}T\right)\right]$, where J is the exchange interaction within the Cu^2+^ dimers, while θ′ is the Weiss constant characterizing the inter-dimer coupling. The dash-dotted line in Figure 4b illustrates the fitting result for the μ_eff_(T) curve (complex **2**) with J/k_B_ ≈ 31 K and θ^′^ ≈ 0.47 K. Somewhat weaker exchange interactions J/k_B_ ≈ 25 K and θ^′^ ≈ 0.30 K were evaluated for complex **1**.

The presence of FM exchange interactions between Cu^2+^ dimers is substantiated by the field dependences of magnetization *M*(*H*) (Figure 5) that demonstrate larger values of magnetization and faster saturation than expected for a set of isolated *S* = 1 moments (dimers in the triplet state). The stronger inter-dimer interaction in complex **2** in comparison with **1** is clearly manifested in Figure 5 by larger *M* values.

In contrast to the first pair of complexes, complexes **3** and **4** demonstrate a behavior much closer to the ideal paramagnetic one (Figure 6). Their magnetic susceptibility can be well described by the Curie–Weiss law over the entire accessible temperature range 1.77–300 K with µ_eff_ ≈ 1.72 µ_B_ and 1.70 µ_B_, θ ≈ −0.4 K and −0.65 K for complexes **3** and **4**, respectively (Figure 6b). Apparently, the exchange interaction between Cu^2+^ ions in complexes **3** and **4** has the antiferromagnetic (AF) sign—opposite to the case of **1** and **2**—and is much weaker than in the latter. The weakness of the interaction in complexes **3** and **4** makes it hardly possible to separate the exchange interactions within Cu^2+^ dimers and between them.

Differences in the magnetic behavior of the two pairs of compounds could be attributed to differences in the geometrical parameters of the dimeric {Cu_2_(μ-X)_2_X_2_} (X = Cl, Br) fragment (Table 1). In particular, in the {Cu_2_(μ-Br)_2_Br_2_} dimer of structure **2**, the Cu-Br(bridging) distances differ little from each other (2.4218(7)–2.4572(7) Å) with almost equal Cu-Br(bridging)-Cu angles (92.493(1) and 93.587(1)°), while in structure **4,** having the same {Cu_2_(μ-Br)_2_Br_2_} unit, the differences in the Cu-Br(bridging) distances and Cu-Br(bridging)-Cu angles are 0.1141 Å and 5.772°.

### 2.6. Oxygenation of Alkanes and Alcohols

We have found that alkanes are oxidized in acetonitrile solution to alkyl hydroperoxides by hydrogen peroxide in air in the presence of catalytic amounts of complexes **1**–**4**. The oxygenation of cyclohexane, methylcyclohexane and *n*-heptane with **1**–**4**–H_2_O_2_ systems under mild conditions (typical temperature 50 °C) has been studied. Curves of accumulation of cyclohexane oxidation products are presented on Figure 7.

All complexes demonstrated similar reactivity. Complex **3** exhibited the highest activity. The kinetic curves of accumulation of cyclohexanol and cyclohexanone measured before and after the addition of PPh_3_ in the cyclohexane oxidation reaction catalyzed by compound **3** are presented in Figure 8. The reaction between an alkane and H_2_O_2_ initially gives the corresponding alkyl hydroperoxide. This peroxide decomposes in a gas chromatograph to form a ketone and an alcohol in comparable amounts (see Figure 8A). Each sample of the reaction solution was analyzed twice before and after batch treatment with excess solid PPh_3_. Triphenylphosphine reduces alkyl hydroperoxide to alcohol. In this regard, the addition of triphenylphosphine to the reaction mixture led to a sharp increase in the concentration of cyclohexanol and a decrease in the concentration of cyclohexanone [84,85], as shown in Figure 8B. In the absence of any catalyst, a maximum yield of cyclohexanol was 0.001 M and no cyclohexanone was detected (after 300 min and addition of PPh_3_).

The yields shown in Figure 7 are comparable to those obtained by the oxidation of alkanes by other copper-based systems. In particular, various types of di-, tri-, tetra- and polymeric copper(II) compounds act as rather efficient catalysts or catalyst precursors in the oxidation of cyclohexane by H_2_O_2_, leading to total product yields in the 22–45% range, with the highest values achieved when using compounds [Cu_2_Co_2_Fe_2_(μ-dea)_6_(NCS)_4_(MeOH)_2_]·3.2H_2_O (45%) [86], [Cu_4_(μ_4_-O)(μ_3_-tea)_4_(μ_3_-BOH)_4_][BF_4_]_2_ (39%) [87], and [(phen)_2_CuCl](PF_6_) (27%) [79].

The selectivity parameters for the oxidation of *n*-heptane and methylcyclohexane with hydrogen peroxide catalyzed by compound **3** were also measured. Regio-selectivity parameter for *n*-heptane oxidation: C(1):C(2):C(3):C(4) = 1.0:5.5:5.7:5.3. The bond-selectivity parameter for the oxidation of methylcyclohexane: 1°:2°:3° = 1.0:7.3:18.0. These parameters are close to the values determined for the oxidation of the corresponding hydrocarbons in systems generating free hydroxyl radicals [88]. Slightly higher values of the parameters in our case can be explained by the shielding of the reaction center by bulky ligands in complex **3**.

Finally, the oxidation of alcohols by *tert*-butyl hydroperoxide has been studied. The accumulation curves of acetophenone during the oxidation of 1-phenylethanol catalyzed by complexes **1** and **3** (yields 90–97%) are shown in Figure 9. The oxidation of cyclohexanol gives lower yields (32–37%) (Figure 10).

## 3. Experimental Section

### 3.1. General Procedures

All manipulations were carried out in air. CuCl_2_·2H_2_O and CuBr_2_ were commercially available. Starting nopinane annelated L_1_ ((1R,3R,8R,10R)-2,2,9,9-Tetramethyl-3,4,7,8,9,10-hexahydro-1H-1,3:8,10-dimethanocyclopenta [1,2-b:5,4-b’]diquinolin-12(2H)-one) and L_2_ ((1R,3R,8R,10R,1’R,3’R,8’R,10’R)-2,2,2’,2’,9,9,9’,9’-Octamethyl-1,1’,2,2’,3,3’,4,4’,7,7’,8,8’,9,9’,10,10’-hexadecahydro-1,3:1’,3’:8,10:8’,10’-tetramethano-12,12’-bi(cyclopenta [1,2-b:5,4-b’]diquinolinylidene)) were synthesized according to the published procedure [35]. All solvents were distilled by standard methods before use.

### 3.2. Physical Measurements

Elemental C, H, and N analyses were performed with a EuroEA3000 Eurovector analyzer. The IR spectra were recorded in the 4000–400 cm^−1^ range with a Perkin–Elmer System 2000 FTIR spectrometer, with samples in KBr pellets and Nujol. EPR spectra were recorded in the X- and Q-bands at 77 and 300 K on an E-109 Varian spectrometer, equipped with an analog-to-digital signal converter. To analyze and simulate EPR spectra, the EasySpin (Matlab software package) was used [89]. All measurements were taken with an external reference DPPH standard (2,2-diphenyl-1-picrylhydrazyl) for the correct determination of g tensor values.

### 3.3. Magnetic Measurements

Magnetic properties of polycrystalline samples were studied using a Quantum Design MPMS-XL SQUID magnetometer in the temperature range of 1.77–300 K at magnetic fields *H* = 0–10 kOe. To check the magneto-thermal reversibility, temperature dependences of the magnetization, *M*(*T*), were subsequently measured on heating the sample after it had been cooled in a zero magnetic field (ZFC) and after cooling in a given magnetic field (FC). To determine the paramagnetic component of the molar magnetic susceptibility *χ*_p_(*T*), the temperature-independent diamagnetic contribution *χ*_d_ and possible contribution of ferromagnetic microimpurities χ_F_ were subtracted from the measured values of the total molar susceptibility *χ* = *M/H*: *χ*_p_(*T,H*) = *χ*(*T,H*) − *χ*_d_ − *χ*_F_(*T,H*). The value of *χ*_d_ was calculated according to the additive Pascal scheme, while the ferromagnetic contribution *χ*_F_, if any, was evaluated from the field dependences *M*(*H*). To determine the effective magnetic moment *µ*_eff_ and the Weiss constant *θ*, the temperature dependences *χ*_p_(*T*) were analyzed using the Curie–Weiss dependence $\chi_{p}\left(T\right)=N_{A}\mu_{\mathrm{eff}}^{2}/3k_{B}\left(T-\theta \right)$, where *N*_A_ and *k*_B_ are the Avogadro number and the Boltzmann constant, respectively.

### 3.4. X-ray Data Collection and Structure Refinement

X-ray diffraction data for **4** were obtained on an Agilent Xcalibur diffractometer equipped with an area CCD AtlasS2 detector (MoKα, λ = 0.71073 Å, graphite monochromator, ω-scans). Integration, absorption correction, and determination of unit cell parameters were performed using the CrysAlisPro program package [90]. Diffraction data for **1** and **2** were collected with a Bruker D8 Venture diffractometer equipped with an area CMOS PHOTON III detector and IµS 3.0 source (Mo Kα, λ = 0.71073 Å, φ- and ω-scan). Absorption corrections were applied with the use of the SADABS program [91]. The structures were solved by the dual space algorithm (SHELXT) [92] and refined by the full-matrix least-squares technique (SHELXL) [93] in the anisotropic approximation (except hydrogen atoms). Positions of hydrogen atoms of organic ligands were calculated geometrically and refined in the riding model. Guest CHCl_3_ molecules in structure **4** were highly disordered and could not be modeled as a set of discrete atomic sites. The final formula of compound **4** was evaluated from the results of the PLATON/SQUEEZE [94] procedure (449 *e*^−^ in 1482 Å^3^). The crystallographic data and details of the structure refinement are summarized in Table 2. Selected bond distances are listed in Table 1. CCDC 2,165,291–2,165,293 contain the crystallographic data for **1**–**2**, **4**, respectively. These data can be obtained free of charge from The Cambridge Crystallographic Data Centre via https://www.ccdc.cam.ac.uk/structures/ accessed on 8 April 2022.

#### 3.4.1. Synthesis of [CuCl_2_L_1_]_2_ (**1**)

A mixture of CuCl_2_•2H_2_O (50 mg, 293 µmol) and dipinodiazafluorenone L_1_ (109 mg, 293 µmol) was dissolved in 10 mL of ethanol and 1 mL of chloroform (L_1_ is poorly soluble in pure ethanol) with stirring and heating (45 °C). After complete dissolution, heating was stopped, and the mixture was continuously stirred at room temperature for a day. The resulting bright green solution was evaporated to dryness and washed with diethyl ether. Crystals were obtained by layering diethyl ether on a solution of **1** in dichloromethane. Green crystals formed after 2 days. Yield: 109 mg (74%). Anal. Calc. for C_25_H_26_Cl_2_CuN_2_O: C 59.5, H 5.2, N 5.5; Found C 59.8, H 4.8, N 5.8. IR (KBr) ν/cm^−1^: 3439 (m), 2957 (s), 2918 (s), 2872 (m), 1726 (s), 1574 (s), 1493 (w), 1468 (m), 1420 (s), 1400 (s), 1352 (m), 1258 (s), 1213 (m), 1182 (w), 1150 (m), 1070 (m), 1053 (m), 947 (m), 897 (w), 826 (w), 799 (s), 785 (s), 743 (m), 602 (w), 497 (w).

#### 3.4.2. Synthesis of [CuBr_2_L_1_]_2_ (**2**)

Dipinodiazafluorenone L_1_ (61.3 mg, 165.5 µmol) was dissolved in 3 mL of chloroform to give a bright orange solution. CuBr_2_ (37.0 mg, 165.6 µmol) was dissolved in 2 mL of acetonitrile to give a dark green solution. Stirring the mixture at room temperature for one day gave a green solution, which was evaporated to dryness. Crystals were obtained by slow diffusion of Et_2_O into a solution of **2** in dichloromethane. Yield: 54 mg (55%). Anal. Calc. for C_25_H_26_Cl_2_CuN_2_O: C 50.6, H 4.4, N 4.7; Found: C 50.6, H 4.5, N 4.8. IR (KBr) ν/cm^−1^: 3441 (s), 2990 (m), 2954 (s), 2918 (s), 2870 (m), 1726 (s), 1574 (s), 1493 (w), 1466 (w), 1420 (m), 1400 (s), 1352 (w), 1258 (s), 1213 (w), 1182 (w), 1150 (m), 1070 (m), 1053 (w), 947 (w), 897 (w), 826 (w), 797 (m), 785 (m), 743 (w), 602 (w), 498 (w).

#### 3.4.3. Synthesis of [(CuCl_2_)_2_L_2_]_n_ (**3**)

Dipinodiazafluorene L_2_ (58 mg, 81.8 μmol) was dissolved in 3 mL of chloroform (bright orange solution). CuCl_2_•2H_2_O (27.9 mg, 163.6 μmol) was dissolved in 2 mL of methanol (bright green solution). Both solutions were mixed. The mixture was stirred at room temperature for 1 h then evaporated to dryness to give a dark brown crystalline powder which was washed with diethyl ether. Yield: 68 mg (73%). Anal. Calc. for C_50_H_52_Cl_4_Cu_2_N_4_•0.5CHCl_3_: C 58.5, H 5.1, N 5.4; Found: C 58.6, H 5.3, N 5.5. IR (KBr) ν/cm^−1^: 3408 (m), 2922 (s), 1726 (m), 1631 (m), 1572 (s), 1497 (m), 1470 (m), 1418 (s), 1398 (s), 1258 (s), 1217 (m), 1190 (m), 1072 (m), 1049 (w), 945 (w), 918 (w), 823 (w), 750 (m).

#### 3.4.4. Synthesis of [(CuBr_2_)_2_L_2_]_n_ (**4**)

A solution of dipinodiazafluorene L_2_ (58 mg, 81.8 µmol) in 3 mL of chloroform was added to a solution of CuBr_2_ (36.5 mg, 163.6 µmol) in 2 mL of acetonitrile. The mixture was stirred at room temperature for 1 h to give a brown solution. The solution was evaporated to dryness to give a dark brown crystalline powder. It was washed with diethyl ether. Crystals suitable for X-ray diffraction analysis were obtained by slow evaporation of the brown solution. Yield: 61 mg (59%). Anal. Calc. for C_50_H_52_Cl_4_Cu_2_N_4_•CHCl_3_: C 48.0, H 4.2, N 4.4; Found: C 47.7, H 4.1, N 4.3. IR (KBr) ν/cm^−1^: 3449 (s), 2972 (m), 2924 (m), 1636 (s), 1501 (w), 1497 (w), 1464 (w), 1416 (m), 1398 (m), 1256 (m), 1219 (w), 1190 (w), 1072 (w).

### 3.5. Catalytic Studies

All reactions were carried out in air in thermostatically controlled cylindrical glass vessels with vigorous stirring. The total volume of the reaction solution was 5 mL (WARNING: the combination of air or molecular oxygen and H_2_O_2_ with organic compounds can be explosive at elevated temperatures!). Initially, a portion of a 50% aqueous hydrogen peroxide solution was added to a solution of the catalyst and substrate in acetonitrile. Aliquots of the reaction solution were analyzed by GC (3700 instrument, FFAP/OV-101 20/80 *w*/*w* fused silica capillary column, 30 m × 0.2 mm × 0.3 μm, argon as a carrier gas). The reactions of alkanes and alcohols were stopped by cooling and were usually analyzed twice, i.e., before and after the addition of excess solid PPh_3_. Nitromethane was used as an internal standard. It was added after PPh_3_ at room temperature.

### 3.6. HR-ESI-MS Studies

The high-resolution electrospray ionization mass spectrometric (HR-ESI-MS) measurements were obtained with a direct injection of liquid samples via automatic syringe on an ESI quadrupole time-of-flight (ESI-Q-TOF) high-resolution mass spectrometer Maxis 4G (Bruker Daltonics, Bremen, Germany). The spectra were recorded in the 300–3000 m/z range in positive mode. Typical resolution of MS spectra was ca. 50,000, accuracy < 1 ppm. Exact masses and isotopic patterns were calculated with Compass IsotopePattern v3.0 software software (Bruker Daltonics, Germany).

## 4. Conclusions

New copper(II) complexes with dipinodiazafluorene ligands **1**–**4** were obtained in yields from 55 to 74% and characterized by IR and EPR spectroscopy, HR-ESI-MS, and elemental analysis. The crystal structures of compounds **1**, **2**, and **4** were determined. Complexes **1**–**2** have an island structure, while complex **4** has a polymeric structure due to additional coordinating N,N donor centres in L_2_. All structures contain a binuclear fragment {Cu_2_(μ-X)_2×2_} (X = Cl, Br). The CuX3N2 coordination site has the geometry of a distorted square pyramid. All copper(II) complexes are paramagnetic (*S* = 1/2 per each Cu^2+^) in accordance with the d^9^ electronic configuration, which was confirmed by EPR spectroscopy and magnetic susceptibility measurements. In monomeric complexes **1** and **2**, intra- and intermolecular ferromagnetic exchange interactions between paramagnetic centers were found. On the contrary, in polymeric compounds **3** and **4**, the magnetic exchange interactions were of a weak antiferromagnetic nature, which is apparently associated with differences in the geometric parameters of the dimeric {Cu_2_(μ-X)_2×2_} (X = Cl, Br) unit. All complexes exhibit high catalytic activity in the oxidation of alkanes (cyclohexane, *n*-heptane, and methylcyclohexane) with hydrogen peroxide and alcohols (phenylethanol and cyclohexanol) with *tert*-butyl hydroperoxide. Measurement of *n*-heptane regioselectivity and bond selectivity in the reactions of methylcyclohexane with H_2_O_2_ allows us to conclude that hydroxyl radicals play an important role in these oxidation reactions. The starting products in reactions with alkanes are alkyl hydroperoxides, which are reduced to the corresponding alcohols with an excess of triphenylphosphine (PPh_3_).

## Data Availability

Not applicable.

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
