# Peer review of "Novel Copper(II) Complexes with Dipinodiazafluorene Ligands: Synthesis, Structure, Magnetic and Catalytic Properties"

_molecules, 2022, doi:10.3390/molecules27134072_

Round 1

Reviewer 1 Report

In the manuscript “Novel copper(II) complexes with dipinodiazafluorene ligands: synthesis, structure, magnetic and catalytic properties” the authors describe the synthesis, structural, spectroscopic, and magnetic properties of four new compounds. The crystal structure of three compounds from the four described is determined by single-crystal X-ray diffraction. The catalytic activity of the compounds was also examined. The scientific contents are finely presented and gave the contribution to chemistry. Nevertheless, there are a couple of problems that need correction to improve the quality of the manuscript. I recommend the publication of this manuscript after the necessary corrections.

11.       The introduction of the manuscript is a bit too concisely written. Namely, the authors wrote about pinanes, pinopyridines, 4,5-diazafluorene, and bistricyclic aromatic enes, but the connection between them is missing. By adding a few sentences in the first part of the introduction it would be clearer and easier for the reader to understand the aim of the paper.

22.       In the discussion are missing the names of the ligands. They should be added at the beginning of the discussion. The abbreviations of the ligands are visually loading the text. They should be changed to simpler ones, e.g. to L2 and L4, or simply L and L’.

33.       In the capture below Figure 1, the и should be changed into and.

44.       In the ESI-MS studies part, the mass fragments are listed. Instead of such a listing, the spectra or at least one spectrum should be added to the manuscript. Besides, in the textual part, the implication deducted from the ESI-MS results should be described.

55.       In the EPR studies part, the figures are not presented in a completely clear way. Namely, in figure 2 on the top figure, the red curve is the experimental one signed as “a”, while on the bottom figure the black curve is the experimental “a” curve.  Instead of “top” and “bottom figure” they should be marked as “2a” and “2b”. The curves should be colored in the same way, with the same color for experimental ones and with the other color for simulated curves. Figure 3 should be corrected analogously.

66.       In the part of magnetic measurements, the equations are written in a larger font than the text generally. In the 8. row of this section phrase “EPR experiments” also stays in a greater font. On figure 4a the blue triangles and red dots are not visible. Why?

77.       In the oxygenation part, the explanation about alkyl hydroperoxide decomposition to ketone and alcohol should be placed at the beginning of this part. The results should be given and discussed after the short explanation of the experiment.

Author Response

In the manuscript “Novel copper(II) complexes with dipinodiazafluorene ligands: synthesis, structure, magnetic and catalytic properties” the authors describe the synthesis, structural, spectroscopic, and magnetic properties of four new compounds. The crystal structure of three compounds from the four described is determined by single-crystal X-ray diffraction. The catalytic activity of the compounds was also examined. The scientific contents are finely presented and gave the contribution to chemistry. Nevertheless, there are a couple of problems that need correction to improve the quality of the manuscript. I recommend the publication of this manuscript after the necessary corrections.

  1. The introduction of the manuscript is a bit too concisely written. Namely, the authors wrote about pinanes, pinopyridines, 4,5-diazafluorene, and bistricyclic aromatic enes, but the connection between them is missing. By adding a few sentences in the first part of the introduction it would be clearer and easier for the reader to understand the aim of the paper.

Response: The introduction has been rewritten.

  1. In the discussion are missing the names of the ligands. They should be added at the beginning of the discussion. The abbreviations of the ligands are visually loading the text. They should be changed to simpler ones, e.g. to L2 and L4, or simply L and L’.

Response: The abbreviations of the ligands LNN and LNNNN have been changed into L1 and L2. Names of the ligands have been added at the beginning of the discussion.

  1. In the capture below Figure 1, the Ð¸should be changed into and.

Response: This has been corrected.

  1. In the ESI-MS studies part, the mass fragments are listed. Instead of such a listing, the spectra or at least one spectrum should be added to the manuscript. Besides, in the textual part, the implication deducted from the ESI-MS results should be described.

Response: The ESI-MS studies part has been rewritten. The conclusion to this part is presented in the last paragraph. All details are given in the Supporting Information. Figures of the mass spectra are given in the Supporting Information to unload the main text of the article.

  1. In the EPR studies part, the figures are not presented in a completely clear way. Namely, in figure 2 on the top figure, the red curve is the experimental one signed as “a”, while on the bottom figure the black curve is the experimental “a” curve.  Instead of “top” and “bottom figure” they should be marked as “2a” and “2b”. The curves should be colored in the same way, with the same color for experimental ones and with the other color for simulated curves. Figure 3 should be corrected analogously.

Response: This has been corrected.

  1. In the part of magnetic measurements, the equations are written in a larger font than the text generally. In the 8. row of this section phrase “EPR experiments” also stays in a greater font. On figure 4a the blue triangles and red dots are not visible. Why?

Response: The equations and phrase “EPR experiments” have been corrected. Triangles and dots are not visible on the Figure 4 since the graphs are almost identical and overlap each other.

  1. In the oxygenation part, the explanation about alkyl hydroperoxide decomposition to ketone and alcohol should be placed at the beginning of this part. The results should be given and discussed after the short explanation of the experiment.

Response: This has been done.

We appreciate the comments of the Reviewer that have helped us to improve the present manuscript and hope that it will be acceptable in the corrected form.

Reviewer 2 Report

The manuscript molecules-1784896 "Novel copper(II) complexes with dipinodiazafluorene ligands: synthesis, structure, magnetic and catalytic properties" by Gushchin and co-workers describes the synthesis of four copper(II) complexes with dipinodiazafluorenes and the study of their structures and physicochemical properties. The synthesis of the obtained complexes was confirmed by IR and EPR spectroscopy, HR-ESI-MS, and elemental analysis. The crystal structure of three complexes was confirmed by single-crystal X-ray diffraction. The authors have interesting experimental results, so I believe that this paper will be of interest to the readers of Molecules.

Questions and comments:

1) The introduction looks short and does not give the reader a complete understanding of the research topic. The manuscript also has high self-citation, i.e. ~53% (27 out of 51 refs). In this regard, I recommend that the authors strengthen the introduction by paying attention not only to their publications, but also to the work of colleagues in the field of research on the synthesis and properties of complexes.

2) How can the authors explain the low yields (55–74%) of the obtained complexes? Has the optimization of the isolation of target complexes been studied?

3) What can be said about the binding constants of the copper cation by the studied ligands? Does the anion affect the efficiency of binding Cu(II) ions?

4) The obtained data on the catalytic activity of copper (II) complexes should be compared with the previously studied analogs known in the literature.

5) Table S1. I recommend using lowercase to write the found form (for example, C25H26N2O) in first column. Please correct the title of the second column (in Russian).

6) Minor changes.

- The list of references does not contain titles of publications and doi.

- Please place the images in the Figure 1 on one page without breaking.

- The manuscript lacks information about the device and the conditions for recording mass spectra.

Author Response

The manuscript molecules-1784896 "Novel copper(II) complexes with dipinodiazafluorene ligands: synthesis, structure, magnetic and catalytic properties" by Gushchin and co-workers describes the synthesis of four copper(II) complexes with dipinodiazafluorenes and the study of their structures and physicochemical properties. The synthesis of the obtained complexes was confirmed by IR and EPR spectroscopy, HR-ESI-MS, and elemental analysis. The crystal structure of three complexes was confirmed by single-crystal X-ray diffraction. The authors have interesting experimental results, so I believe that this paper will be of interest to the readers of Molecules.

  1. The introduction looks short and does not give the reader a complete understanding of the research topic. The manuscript also has high self-citation, i.e. ~53% (27 out of 51 refs). In this regard, I recommend that the authors strengthen the introduction by paying attention not only to their publications, but also to the work of colleagues in the field of research on the synthesis and properties of complexes.

Response: The introduction has been reworked, new references have been added.

  1. How can the authors explain the low yields (55–74%) of the obtained complexes? Has the optimization of the isolation of target complexes been studied?

Response: Yields are given for fine crystalline products. Synthesis procedures were optimized to obtain pure phases. 

  1. What can be said about the binding constants of the copper cation by the studied ligands? Does the anion affect the efficiency of binding Cu(II) ions?

Response:  Binding constants have not been studied. We assume that they should be close to the binding constants of the copper(II) cation and bipyridyl ligands. Anions should not significantly affect the efficiency of binding Cu(II) ions?

  1. The obtained data on the catalytic activity of copper (II) complexes should be compared with the previously studied analogs known in the literature.

Response: This has been done.

  1. Table S1. I recommend using lowercase to write the found form (for example, C25H26N2O) in first column. Please correct the title of the second column (in Russian). 

Response: This has been corrected.

  1. Minor changes.

The list of references does not contain titles of publications and doi. (Fomenko)

Response: This has been corrected.

Please place the images in the Figure 1 on one page without breaking. (Fomenko)

Response: This has been corrected.

The manuscript lacks information about the device and the conditions for recording mass spectra.

Response: This information has been added (Part 3.6. HR-ESI-MS studies).

We appreciate the comments of the Reviewer that have helped us to improve the present manuscript and hope that it will be acceptable in the corrected form.

Reviewer 3 Report

Fomenko presented a detailed study of four binuclear Cu(II) complexes. The structures of those complexes are well characterized by single-crystal XRD, ESR, SQUID, and MS. They also showed that those complexes have catalytic activity in the oxidation of alkanes.

The following points need to be addressed before publication,

1.       The synthesis procedure of ligand LNN and LNNN should be included.

2.       The blank experiment without adding catalyst should be performed to gauge the background conversion without catalyst but with H2O2 added.

Author Response

Fomenko presented a detailed study of four binuclear Cu(II) complexes. The structures of those complexes are well characterized by single-crystal XRD, ESR, SQUID, and MS. They also showed that those complexes have catalytic activity in the oxidation of alkanes.

The following points need to be addressed before publication.

  1. The synthesis procedure of ligand LNN and LNNN should be included.

Response: In the experimental part, we refer to work on the synthesis of these compounds (Reference 35).

  1. The blank experiment without adding catalyst should be performed to gauge the background conversion without catalyst but with H2O2 added.

Response: In the absence of any catalyst a maximum yield of cyclohexanol was 0.001 M and no cyclohexanone was detected (after 300 minutes and addition of PPh3).

We appreciate the comments of the Reviewer that have helped us to improve the present manuscript and hope that it will be acceptable in the corrected form.

Round 2

Reviewer 1 Report

The manuscript is improved in accordance with my comments. Only a few spelling mistakes need corrections and it can be accepted for publication.

Please correct on page 10/24, last paragraph on the page: instead of "alkanes by other cupper based systems" should stay "alkanes by other copper based systems".

From page 15/24, part 3.4.1. Synthesis of [CuCl2L1]2 (1): The center dot in the formula CuCl2•2H2O should be replaced by CuCl2∙2H2O.

The text after reference brackets starting from [84] should be removed.

Reviewer 2 Report

I thank the authors for answering my questions and improving the manuscript.